# Effective Unlearning in LLMs Relies on the Right Data Retention Strategy

## Abstract

Unlearning in Large Language Models (LLMs) has gained increasing attention in recent years due to its critical role in ensuring ethical and legal compliance. Although significant progress has been made in developing unlearning algorithms, relatively little attention has been devoted to the *data perspective*. In particular, the role of retain-set selection in preserving *model utility* remains underexplored, even though it is critical for making unlearning practical in real-world applications. In this work, we explore strategies for constructing effective retain sets by adapting methods from coreset selection and prior unlearning research. We evaluate these approaches on two complementary datasets: *(i)* a monotonic dataset built from a benchmark dataset, and *(ii)* a mixed, larger-scale dataset combining WPU, TOFU, and Dolly, which better reflects realistic scenarios where forget and retain samples are not explicitly defined. We find that *model utility* is strongly influenced by the model's representations within the selected retain set for heterogeneous dataset. Moreover, we show that simply choosing data samples with high semantic or syntactic similarity to the forget set can yield substantially better results than standard coreset techniques. To the best of our knowledge, this work represents the first systematic study of practical retain-set selection for LLM unlearning, highlighting its importance and the challenges it poses in practical settings.

## 1 Introduction

Large Language Models (LLMs) (Vaswani et al., 2017), with their remarkable capabilities across a wide range of tasks and training on vast amounts of web data, inevitably face alignment challenges. These models often memorize undesirable information (Carlini et al., 2021; Golatkar et al., 2020) such as personal data, copyrighted material, and harmful content which can be outputted and potentially misused (Staab et al., 2024). Alignment techniques, such as Reinforcement Learning from Human Feedback (RLHF) (Ouyang et al., 2022) and red teaming, have been introduced to mitigate these risks, but they require substantial human effort. Furthermore, these methods do not fully address legal requirements, such as the GDPR's *Right to be Forgotten* or the AI Act. Machine unlearning (Yao et al., 2024; Miranda et al., 2025) has emerged as a promising alternative, aiming to remove specific undesired information and abilities while preserving overall model utility.

LLM unlearning generally has two key objectives: (1) eliminating the specified target knowledge along with its associated capabilities, and (2) preserving the model's overall integrity by preventing degradation of non-target knowledge and abilities Liu et al. (2025). Achieving these objectives at the same time, requires two datasets: the *forget set $D_f$*, containing the data to be removed, and the *retain set $D_r$*, containing the knowledge to be preserved. By definition $D_r$ and $D_f$ are disjoint, and together they cover the complete corpus $D$, i.e., $D_r = D \setminus D_f$ and $D_f = D \setminus D_r$. Since $D_f$ is usually smaller than $D_r$, there is often a disproportionality in the dataset sizes. Mainstream unlearning approaches (Zhang et al., 2024; Yuan et al., 2025; Maini et al., 2024; Liu et al., 2022; Jang et al., 2023) address the two objectives through a weighted combination of losses: maximizing the forget loss on $D_f$ while minimizing the retain loss on $D_r$, typically formulated as (Ji et al., 2024).

$$\min_{\theta} \ \mathbb{E}_{(x,y)\in \mathcal{D}_f}[\ell_f(y \mid x; \theta)] + \lambda \, \mathbb{E}_{(x,y)\in \mathcal{D}_r}[\ell_r(y \mid x; \theta)] , \tag{1}$$

$\theta$ denotes the model parameters subject to update during unlearning, initialized from the pretrained model. The terms $\mathcal{L}_f$ and $\mathcal{L}_r$ denote the forget loss (unlearning objective) and the retain loss (utility-preserving objective), respectively. Both are evaluated when generating a response $y$ from an input $x$ under parameters $\theta$. The coefficient $\lambda > 0$ functions as a regularization parameter, balancing the trade-off between forgetting and retention. Based on the above, unlearning performance is measured along the two objectives, **Forget Quality (FQ)** focuses on objective (1) capturing how effectively the model forgets the targeted knowledge, i.e., the extent to which the undesired information is no longer recoverable through direct or indirect queries. In contrast, **Model Utility (MU)** focuses on objective (2), reflecting how well the model retains its general capabilities, ensuring that unlearning does not significantly impair its performance on unrelated tasks or knowledge domains. *FQ* is applied on $D_f$ and *MU* applied on $D_r$.

Early unlearning methods (Yao et al., 2024; Zhang et al., 2024) focused solely on maximizing the loss on $D_f$, often leading to **degeneration behavior** and **catastrophic forgetting** (Jang et al., 2023; Ji et al., 2024). $D_r$ was later introduced as a regularization set to mitigate these issues. The retain loss $\mathcal{L}_r$ is typically defined as the standard cross-entropy next-token prediction loss computed over $D_r$. Much of the research in LLM unlearning has focused on developing algorithms to balance the forgetting and retention objectives. Prominent approaches include: (i) Gradient Ascent and its variants (Jin et al., 2025; Wang et al., 2025), which reverse the training loss to enforce forgetting; (ii) preference optimization methods such as DPO (Rafailov et al., 2023) and its extensions NPO (Zhang et al., 2024) and SimNPO (Fan et al., 2024), which indirectly bound the forgetting objective by increasing preference for desired responses; (iii) representation misdirection techniques such as RMU (Li et al., 2024), which disrupt internal representations tied to the undesired knowledge; (iv) logit-based methods that leverage auxiliary models to reduce preference for $D_f$ (Ji et al., 2024); and (v) model-editing strategies employing task vectors or surgical weight modifications to remove specific knowledge (Wu et al., 2023; Jia et al., 2024; Hase et al., 2023). Across these approaches, $D_r$ is incorporated either directly in the loss function or as part of separate training stages to mitigate catastrophic forgetting.

Although these algorithmic advances are significant, they have been primarily evaluated on benchmark datasets that are relatively simple, i.e., monotonic in structure, and offering a clear separation between forget and retain sets. In addition to this, they typically rely on the full retain set provided. In contrast, real-world scenarios are far more complex: the pre-training dataset $D$ may span gigabytes of data and contain hundreds of thousands of samples, especially in specialized domains such as law or medicine, making it impractical to use the entire dataset (excluding $D_f$) as the retain set. This challenge motivates the question: **"How can we select a subset $D_s$ from $D_r$ that faithfully reflects $D_r$ while preserving model utility?"** Recent works (Ren et al., 2025; Geng et al., 2025) have raised this question but no comprehensive methodology has been established to address it.

In this work, we address the bottleneck problem of retain set selection from a *pre-unlearning* perspective. We perform *early selection* of $D_r$. We draw on established research of *"coreset"* selection methods, adapting it to the unlearning domain and conduct extensive empirical studies on these selected retain sets and on their impact. More specifically, we investigate which samples are selected for retention and examine which properties of the retained data influence model utility.

Our analysis indicates a key pattern: a statistically significant negative correlation between the variance of the model's hidden state representations (hidden state variance, HSV) for data-points in the selected retain set and the model's overall utility, suggesting that higher variance in retained data can reduce model utiliy; In other words, unlearning with widely distributed retain data points tends to reduce model utility. Building on this insight, and informed by prior work showing that syntactically similar samples are most affected during unlearning (Chang & Lee, 2025), we propose two simple selection strategies: retaining semantically closer samples and retaining syntactically closer samples. We then perform extensive empirical studies on $D_r$ using coreset-based methods.

## 2 RELATED WORK

### 2.1 DATA SELECTION

Data selection involves choosing a subset of data from a larger dataset to train machine learning models efficiently. These methods aim to reduce computational costs without compromising per-

formance. Typical paradigms range from heuristic-based selection (e.g., statistical properties, distances) to optimization-based methods (e.g., ranking samples based on loss values, gradients, or forgetting events). Common goals in data selection include **distribution matching** and **distribution diversification** (Albalak et al., 2024).

### 2.1.1 DATA SELECTION IN LARGE LANGUAGE MODELS

In LLMs, data selection is critical for achieving state-of-the-art performance across tasks such as reasoning, instruction tuning, and alignment (e.g., Deepseek V3 (DeepSeek-AI et al., 2025), WizardLM (Xu et al., 2025), Vicuna (Peng et al., 2023), Zephyr (Tunstall et al., 2023)). Typical pipelines include filtering (e.g., language, toxicity, PII), de-duplication, and data mixing. Instruction-tuning datasets often exploit larger models (e.g., GPT-4) for sample annotation, as in DEITA (Liu et al., 2024a), Instag (Lu et al., 2024), and AlpagaSus (Chen et al., 2024), which assess sample complexity, diversity, and quality. While effective, these methods still require costly tagging and pre-selection procedures.

### 2.1.2 CORESET SELECTION

Coreset selection aims to identify a representative subset of data that preserves key distributional properties while maintaining near full-data performance. Approaches in the literature assign importance scores to samples using training dynamics (e.g., gradient norm, error vector norms, forgetting scores) (Paul et al., 2021; Toneva et al., 2019) or emphasize diversity through clustering distances and coverage criteria (Xia et al., 2023; Zheng et al., 2023). Optimization-based methods leverage gradient information to construct subsets (Mirzasoleiman et al., 2020; Killamsetty et al., 2021; Pooladzandi et al., 2022). In LLMs, sample influence can be estimated via gradient similarity with validation data (Xia et al., 2024).

Recently, coreset selection has been applied to unlearning: Patil et al. (2025) prune forget sets using anomaly detection on hidden representations, balancing forgetting and utility preservation, while Pal et al. (2025) investigate underlying coreset behaviour (in $D_f$) in LLM unlearning benchmarks.

### 2.1.3 LLM UNLEARNING AND DATA PERSPECTIVES

Dynamic unlearning methods (Bărbulescu & Triantafillou, 2024) iteratively select highly memorized forget-set samples, and gradient-based approaches (Tian et al., 2024) target sensitive parameters for unlearning. While these methods focus on the model perspective, they highlight the importance of data selection. From a retain-set perspective, Chang & Lee (2025) show syntactic neighbors are highly influential and should be included in benchmark datasets. Bushipaka et al. (2025) explore constructing $D_r$ with multiple neighbors for benchmarks.

Compared with existing studies, our approach differs in two key aspects. First, we select $D_r$ using coreset mechanisms in realistic unlearning scenarios rather than relying on neighbor-based constructions for benchmark datasets. Second, coreset selection does not require well-maintained datasets to identify syntactic or semantic relationships, making it more practical for real-world applications.

### 2.1.4 IMPORTANCE OF RETAIN SET

(Ko et al., 2024)'s work on text-to-image diffusion model unlearning shows that unlearning without a diverse $D_r$ leads to degraded image quality and poor text-image alignment, showing how retain data stabilizes model outputs when concepts are removed. In LLMs, (Thaker et al., 2025) show that narrowly defined forget and retain sets lead unlearning to overfit on the test queries. Beyond MU preservation, $D_r$ is also used in adversarial attacks on the forgotten samples. For instance, (Łucki et al., 2024) find that an unlearned model via RMU shows a significant drop in FE, when finetuned with just 5 unrelated samples from the $D_r$.

## 3 METHOD

### 3.1 PROBLEM STATEMENT

Assume we have a downstream task LLM, instruction-tuned on general knowledge, which has been further fine-tuned on a dataset $D$ containing undesired knowledge that must be removed. Our objective is to select a subset of the retain set, $D_s \subset D_r$, that preserves the model's knowledge and capabilities as they were prior to unlearning.

Here, $D_r = D \setminus D_f$ is the retain set, and a sample $s_i = (x_r, y_r) \in D_r$ is selected if $s_i \in D_s$. The final subset $D_s$ should be substantially smaller than $D_r$ yet sufficient to maintain model utility. In our experiments, we focus on **entity-level unlearning**, removing all knowledge related to a specific concept or individual. Figure 1 illustrates this process.

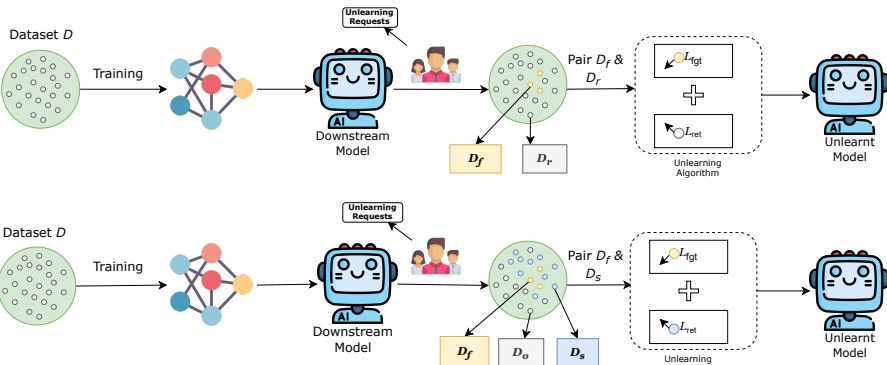

Figure 1: Top: the retain set $D_r$ is traditionally the full dataset $D$ minus the forget set $D_f$. Bottom: our goal is to select a smaller subset $D_s \subset D_r$ that effectively represents $D$ and preserves knowledge after unlearning.

### 3.2 CORESET-BASED DATA SELECTION FOR UNLEARNING

To construct a reliable $D_r$ in practical unlearning scenarios, we employ a rich set of coreset selection methods, specifically **EL2N** and **MODERATE**. We observe a strong correlation between selected $D_s$ hidden state representation variance and model utility.

Additionally, we perform two alternative $D_s$ selections:

1. Greedily selecting the top-n semantically closest samples to each forget sample.
2. Greedily selecting the top-n syntactically closest samples to each forget sample.

These strategies aim to maintain model knowledge while supporting efficient unlearning.

## 4 EXPERIMENTAL SETUP

We conduct entity-level unlearning experiments across two data regimes:

**(a) Monotonic Dataset:** We use the Wikipedia Person Unlearn (WPU) dataset (Liu et al., 2024b), which consists of 100 entities along with their corresponding question-answer pairs extracted from Wikipedia. This dataset follows a monotonic template, similar to other benchmark datasets such as (Maini et al., 2024; Jin et al., 2024), providing Forget-Retain samples in a structured format covering attributes like birthplace, profession, and other factual details.

**(b) Mixed Dataset:** To evaluate performance in a more heterogeneous setting, we combine WPU with TOFU (Maini et al., 2024) and DOLLY (Ouyang et al., 2022). This mixed dataset introduces diversity in content and format, better reflecting real-world unlearning scenarios.

Overall, this setting provides us with a realistic scenario where Unlearning has to be done in a downstream task such as generalized instruction tuning. The mixed dataset contains approximately 21k samples, with the majority sourced from DOLLY. For selecting Forget samples, we follow the splits provided by WPU (Liu et al., 2024b), namely (2, 20, 100) entities. Specifically, for the monotonic dataset, we designate $D_f = 2$ entities for unlearning. For the mixed dataset, we use $D_f = 20$ entities from WPU as the forget set.

Assessing MU across the full 21k samples is computationally expensive and unrealistic. Therefore, we create a test set that includes representative portions from each dataset to enable efficient evaluation. Further details on dataset construction and splits are provided in Appendix A.1.

## 4.1 Unlearning Setup

Our experiments require a retain dataset for regularization. Accordingly, we focus on fine-tuning-based algorithms across three different paradigms. We employ **Gradient Difference** (Liu et al., 2022), **Simple Negative Preference Optimization** (Fan et al., 2024) and **Representation Misdirection Unlearning** (Li et al., 2024) for our experiments.

We do not perform vanilla unlearning (i.e., excluding the retain set) and always include the retain set for regularization. Unlike common practice, which randomly selects retain samples equal in number to forget samples for each epoch (Maini et al., 2024; Liu et al., 2024b; Yuan et al., 2025), we adopt a **Cyclic** setup (Jang et al., 2023). In this setup, $D_f$ is repeatedly cycled until all $D_s$ samples are paired with one forget sample during unlearning. Under the standard implementation (Maini et al., 2024), each epoch uses a retain batch matching the size of the forget batch, with retain examples randomly sampled from the $D_r$. The cyclic approach has been shown to outperform the standard implementation (Premptis et al., 2025; Bushipaka et al., 2025), though it is computationally more expensive. In our setup, it is important to note that dataset partitions vary in cardinality, which implies that the number of steps per epoch is not constant across the 5%, 10%, and 20% splits.

## 4.2 Metrics

Unlearning behavior is best assessed using multiple complementary metrics. We employ a stack of metrics and aggregate them into two scores: **Forget Quality (FQ)** and **Model Utility (MU)**.

The individual metrics are as follows:

- **ROUGE-L**: measures verbatim memorization via word-level overlap.
- **Conditional Probability**: likelihood of the ground-truth answer.
- **Truth Ratio**: likelihood of choosing the correct answer over an incorrect one.[1]
- **Cosine Similarity**: measures semantic similarity in the embedding space.

Following Yuan et al. (2025); Maini et al. (2024), we compute *Forget Quality (FQ)* as *1-Arithmetic mean* of ROUGE-L, Conditional Probability, and Truth Ratio on $D_f$ excluding Cosine Similarity. where Cosine Similarity is excluded for robustness[2].

For *Model Utility (MU)*, we calculate the harmonic mean of ROUGE-L, Conditional Probability, and Cosine Similarity on $D_r$, excluding Truth Ratio.

Our primary focus is on preserving model utility, which requires analyzing the drop in MU before and after unlearning. To quantify this, we follow Chang & Lee (2025) and compute the *Relative Utility Drop (RUD)*:

$$\text{RUD} = \frac{\text{MU}_{\text{pre}} - \text{MU}_{\text{post}}}{\text{MU}_{\text{pre}}} \times 100$$

where $\text{MU}_{\text{pre}}$ and $\text{MU}_{\text{post}}$ denote model utility before and after unlearning, respectively.

---

[1]In contrast to Maini et al. (2024), we do not adopt the p-value from the Kolmogorov–Smirnov test as FQ, since our setting does not allow comparison with a perfectly unlearned model—something even less feasible in real-world scenarios.

[2]We observe that embedding models may return non-zero similarity scores even for nonsensical generations (e.g., continuous dots).

Table 1: Baseline performance on WPU and Mix datasets before and after unlearning (using the full retain set $D_r$) for GradDiff, SimNPO, and RMU.

| Method | WPU | | Mix | |
|---|---|---|---|---|
| | FQ | MU | FQ | MU |
| Pre-unlearning | 0.17 | **0.97** | 0.30 | *0.75* |
| GradDiff | 0.90 | 0.92 | 0.94 | 0.65 |
| SimNPO | 0.80 | *0.94* | 0.84 | **0.76** |
| RMU | 0.89 | 0.55 | 0.93 | 0.46 |

Further details about the metrics are provided in Appendix A.5.

### 4.3 CORESET METHODS

For our experiments, we evaluate three data selection strategies: **RANDOM**, **MODERATE** (Xia et al., 2023), and **EL2N** (Paul et al., 2021). Both MODERATE and EL2N were originally developed for computer vision classification tasks, but have been applied to LLM unlearning by Pal et al. (2025), from whom we gather the implementation procedure.

In particular, EL2N requires an initial warm-up run for a few epochs with the desired loss function. Since unlearning algorithms operate by manipulating loss functions, extracting a coreset with EL2N necessitates running the unlearning loss for several steps. To test whether this warm-up step can be simplified, we additionally experiment with using the standard cross-entropy loss during coreset extraction.

### 4.4 EXPERIMENTAL SETTING

We use the LLaMA 3.1 8B Instruct model (Grattafiori et al., 2024) as our base LLM. Both fine-tuning on the datasets and unlearning are performed using **LoRA** (Hu et al., 2022). All experiments are conducted on a single 40 GB A100 GPU. More details in Appendix:A.2.

## 5 RESULTS

We conduct experiments using three splits of the retain set, $D_r$, corresponding to **5%, 10%, and 20%**[3] for the selection of $D_s$. To maintain comparable evaluation across splits, we filter unlearnt models using method-specific *Forget Quality (FQ)* criteria: GradDiff models with FQ > 0.90 SimNPO with FQ > 0.85 and RMU with FQ > 0.90. The number of training epochs is treated as a hyperparameter to reach this FQ threshold for all splits. This is crucial, as we are looking into preserving the *Model Utility*, when Forgetting is implemented successfully.

Note that we aim to recover the pre-unlearning MU, which is 0.97 for WPU and 0.75 for Mix. Using the full retain set $D_r$ typically preserves MU closer to its pre-unlearning level. For the Mix dataset, we evaluate MU on the constructed test set. The results of these baseline runs are reported in Table 1.

### 5.0.1 PERFORMANCE WITH FULL-RETAIN (BASELINE UNLEARNING)

Table 1 shows the baseline utility on WPU and Mix using the full retain set before and after unlearning. All three unlearning methods introduce a measurable degradation relative to the pre-unlearning model, but the magnitude varies substantially. **SimNPO** preserves utility the best across splits, matching or exceeding pre-unlearning MU on Mix and remaining close on WPU. **GradDiff** induces moderate degradation, with utility reductions on both datasets but still performing competitively in MU. In contrast, **RMU** exhibits the largest utility drop, especially in MU (0.55 on WPU vs. 0.97 pre-unlearning), indicating instability even when the full retain set is available.

---

[3]Initial experiments with 1% and 2% did not yield meaningful results and were therefore discarded.

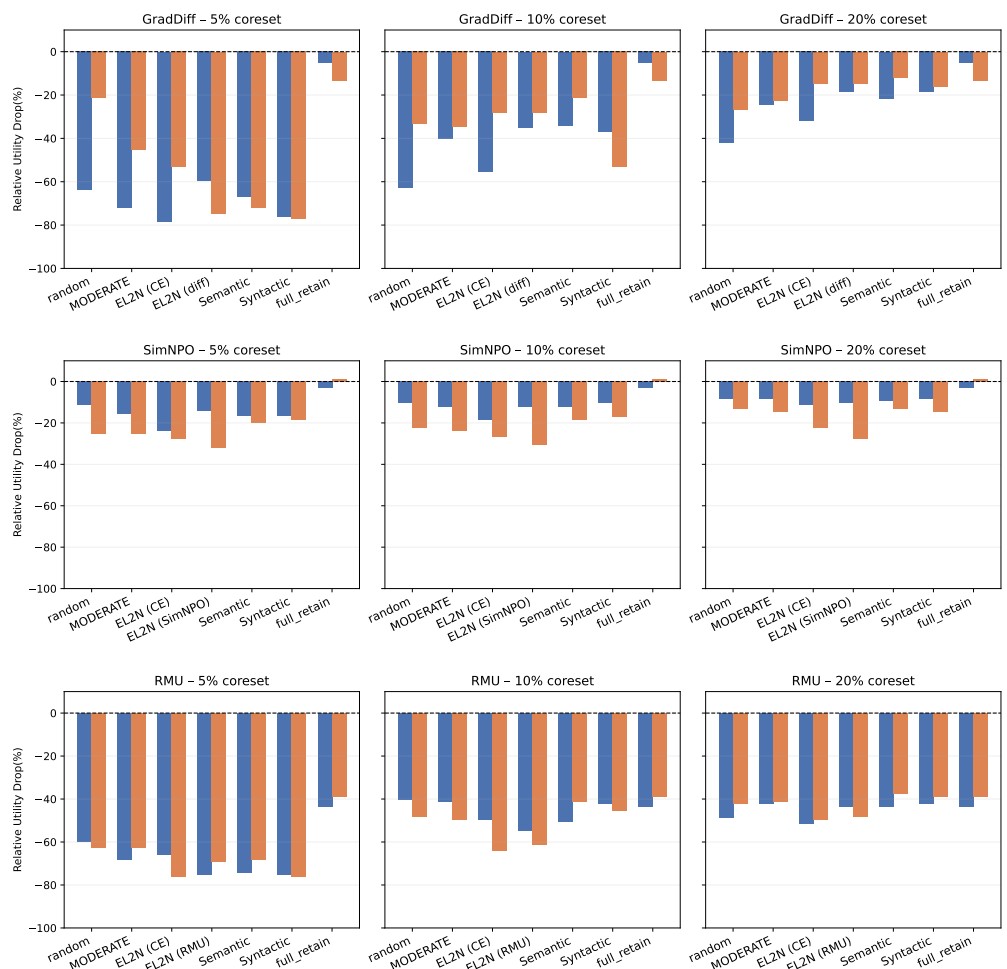

Figure 2: RUD scores for ■WPU ■Mix across 5,10,20% datasets for three unlearning algorithms. We find that targeted selection with semantic and syntactic consistently performs better than other selection mechanisms. However, full retain performs the best and often better than pre-unlearning (SimNPO) showing overfitting of retain set.

### 5.0.2 RELATIVE UTILITY DROP WITH CORESETS

Figure 2 reports the relative utility loss when using coreset subsets of size 5%, 10%, and 20%. Across all methods, **WPU consistently exhibits larger utility drops than Mix**, confirming it is the harder split to preserve performance on. For **Gradient Diff**, utility is highly sensitive to coreset size and selection strategy. At 5%, drops are large (60–80% on WPU), but increasing the coreset size substantially improves stability. At 20%, informed selection strategies—especially **EL2N (diff)**, **Semantic**, and **Syntactic** achieve the best performance, reducing Mix utility drop to as low as 12%. **SimNPO** shows the most stable behavior: utility drops are substantially smaller (often 8–13% at 20%) and vary minimally across selection methods. Syntactic and Semantic coresets consistently perform well, closely matching random sampling at larger sizes. For **RMU**, utility drops remain high across all coreset sizes and strategies, with limited improvement even at 20%. Even the best-performing subsets (Semantic, Syntactic) only reduce the Mix drop to 37–39%, while WPU remains above 42%, signaling persistent sensitivity to data reduction.

Finally, we examine the cluster-level distribution of selected retain samples. Figure 3 shows the log-preference ratio (logPref) for each cluster in the Mix (left) and WPU (right) datasets. Clusters with $\log\text{Pref}(c) \geq 1$ are over-represented (red), $\log\text{Pref}(c) \doteq 0$ are neutral (grey), and $\log\text{Pref}(c) \leq 1$ are

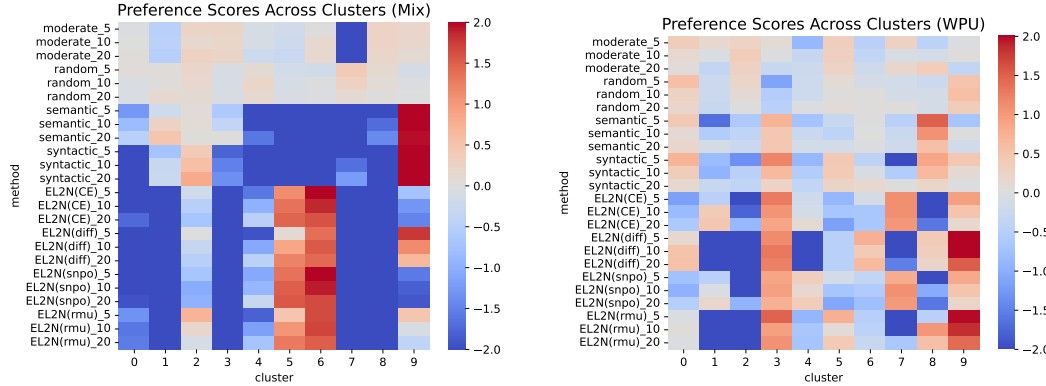

Figure 3: Log-preference ratio (logPref) for clusters in Mix (left) and WPU (right). Clusters with $\text{logPref}(c) \geq 1$ are over-represented (red), $\text{logPref}(c) \doteq 0$ neutral (grey), and $\text{logPref}(c) \leq 1$ under-represented (blue).

under-represented (blue). This visualization allows us to compare how different selection methods distribute samples across clusters without interpreting their effect on utility or forgetting.

## 6 DISCUSSION

Our experimental results reveal important insights regarding data selection strategies for unlearning in LLMs. In the following, we analyze the effectiveness of different coreset methods, examine per-source behavior in heterogeneous datasets, investigate the role of hidden state variance, explore semantic and syntactic selection strategies, and study the impact of cluster-level preferences on model utility and forgetting performance.

**Effectiveness of Coreset Methods and Unlearning Algorithms.** Across both the full-retain–set and coreset experiments, several consistent patterns emerge. First, SimNPO is the most robust and utility-preserving method, both when using the full retain set and when approximating it via a coreset. Its relative utility drop remains small even with only 5–10% of the data, and it shows little sensitivity to the choice of selection heuristic. This suggests that SimNPO's update rule is inherently stable to data sparsification.

Second, Gradient Diff benefits the most from high-quality selection signals. Its performance varies widely depending on coreset choice, especially at small sizes. However, when paired with targeted selection strategies like EL2N (diff) or semantics-driven sampling, GradDiff approaches low relative utility loss at larger coreset sizes, indicating that the gradients it relies on can be well-approximated by carefully chosen examples.

Third, RMU remains the least stable method, showing substantial degradation even with the full retain set and only marginal improvements from any selection strategy. The method appears fundamentally sensitive to removing data from the retain distribution, suggesting its objective does not generalize well under distribution thinning.

Finally, the results highlight a dataset-level trend: WPU is uniformly harder to maintain utility on than Mix, with larger degradations across all methods, coresets, and sizes. This indicates that WPU's distribution has higher dependency on the retained samples, making it more susceptible to information loss during unlearning.

**Hidden State Variance as a Heuristic.** Following (Skean et al., 2025; Tang & Yang, 2025; Duan et al., 2024) we use the last token penultimate layer hidden state representations for our analysis. We observe that Hidden State Variance (HSV) of $D_s$ correlates strongly with Relative Utility Drop (RUD) across all the three unlearning algorithms with GradDiff ($\rho = 0.71, p = 0.01$), RMU ($\rho = $

0.62, p = 0.03) and SimNPO ($\rho = 0.7, p = 0.01$). In contrast, HSV correlates with Forget Quality only for GradDiff ($\rho = 0.64, p = 0.02$), with no significant relationship for RMU or SimNPO. This suggests that Variance in $D_s$ consistently amplifies utility degradation, and in case of GradDiff, also improves the forgetting performance, revealing a method specific trade off. and moderately with Forget Quality on the Mix dataset, but not on WPU. This suggests that higher variance in $D_s$ leads to greater utility drop but also facilitates forgetting, highlighting a trade-off.

Additionally, we conducted a controlled experiment in which we partitioned the full retain set into three subsets based on HSV: low, medium, and high. Using the 10% split, we ran Unlearning experiments across all the three algorithms and observed a consistent pattern: the medium-variance subset yielded the best performance, followed by the high-variance subset, with the low-variance subset performing the worst. Together with the correlation results, this experiment suggests that HSV does not exert a purely linear effect on unlearning, instead performance peaks at intermediate variance, indicating a non-linear, possibly inverted-U relationship. Apart from this, we also find the expected correlations such as RUD negatively correlated to Retain length and FQ. More details in Appendix:7.

**Semantic and Syntactic Selection.** Semantic and syntactic selection methods are more effective than coreset-based approaches, especially on the Mix dataset. Semantic similarity, in particular with GradDiff, achieves comparable performance to full-retain with only 20% $D_s$. However, when the full_retain is unlearnt for *same number of epochs* as semantic, it performs substantially better (2% RUD). The downside is computational cost: full-retain requires 13 hours, whereas semantic method finishes just under 2 hours. This demonstrates that semantic selection can recover approximately 85% of pre-unlearning utility while using an order of magnitude less compute. Finally, we note that relying on full-retain undermines the purpose of unlearning, since it effectively amounts to retraining or fine-tuning the model on the entire $D_r$ again excluding $D_f$.

**Cluster Preferences.** Cluster-level analysis shows that semantic and syntactic methods disproportionately select samples from clusters heavily populated by forget samples (e.g., clusters 9 and 2 in Mix). This selective over-representation explains their effectiveness, while coreset methods—which aim to diversify—suffer higher utility drop A.6.3.

## 7 CONCLUSION

In this study, we address a key limitation in LLM unlearning: the selection of a retain set that preserves Model Utility. We leverage techniques from the coreset selection literature and apply them to entity-level unlearning, evaluating performance on two data regimes: a monotonic dataset (WPU) and a diverse, mixed dataset (Mix). Across both regimes, we find that it is challenging to fully recover the pre-unlearning Model Utility. For the monotonic dataset, the selected subset $D_s$ shows no significant correlation with utility, indicating that standard coreset strategies may not be informative in highly structured or homogeneous data regimes.

In contrast, for the mixed dataset, hidden state variance (HSV) analysis reveals that increasing $Var(D_s)$ leads to a higher Relative Utility Drop (RUD) but also improves Forget Quality. Motivated by this, we implement simple semantic and syntactic-based selection strategies that choose top samples most similar to each forget sample. These approaches consistently outperform traditional coreset methods and, in some cases, even exceed the performance of using the full retain set, demonstrating that targeted retain set selection based on embedding proximity can effectively balance utility preservation and forgetting. Cluster-level analysis further indicates that these methods preferentially select samples from clusters containing forget data, highlighting the importance of considering both dataset structure and relationships between retain and forget samples.

Our findings suggest that while coreset methods provide a strong starting point, understanding the distribution of forget samples in the embedding space and leveraging semantic or syntactic proximity can lead to superior results, especially in heterogeneous datasets, which can be considered as a proxy of real-world scenarios. However, we acknowledge that the observed behavior may not generalize to all unlearning scenarios, such as those involving copyrighted, or harmful content, where additional constraints and safeguards may be required.

In other words, while our study demonstrates the effectiveness of semantic and syntactic-based retain set selection, several avenues remain for future exploration. First, extending these techniques to handle unlearning requests beyond entity-level data—such as instance-level privacy sensitive, copyrighted, or harmful content—would test their generality and robustness. Second, adaptive or dynamic selection strategies that take into account model feedback or embedding evolution during fine-tuning could further improve the trade-off between Model Utility and Forget Quality. Finally, evaluating these methods on larger-scale, multi-domain LLM benchmarks and integrating interpretability or explainability techniques may provide additional insights into why certain selections succeed and inform best practices for practical LLM unlearning.

## 8 LIMITATIONS

LLM unlearning is inherently a dynamic process, requiring continual updates to the model. In contrast, most existing data selection methods are designed as one-time procedures, often involving computationally expensive setups performed prior to training. These static methods are not directly suited for unlearning and would require adaptation to accommodate continuous model updates. Additionally, existing selection strategies are optimized for diversity, ensuring broad dataset coverage, but unlearning requests, particularly entity-level privacy may instead involve densely clustered or non diverse samples. Our experiments show that despite constructing a mixed dataset, the post-training hidden state representations of $D_f$ tend to cluster closely, making diversity oriented selection mechanisms less effective in this context.

As mentioned above, Unlearning is a dynamic process and requires continuous recycling of the model. Often LLM Unlearning is tested in Sequential Setup and found that it is more effective than batch Unlearning. We did not test this setting and would be testing in the later works.

while we show that 20% of the retain set ($D_r$) is sufficient to achieve model utility comparable to that of the full $D_r$, this proportion becomes impractically large for large-scale datasets (e.g., 800k samples). In such scenarios, allocating 20% of the data to unlearn only a small forget set (e.g., 100 samples) is inefficient. Future work should therefore focus on identifying and selecting the most relevant subset of $D_r$, which could substantially reduce this overhead while maintaining unlearning effectiveness. Our work uses only a single Unlearning method and also only one regularization method. Additionally, our experiments are conducted only on single LLM and on only two data regimes. We acknowledge that Unlearning requests can often be varied, such as in Privacy or Copyright contexts. These scenarios require a robust testing of various use cases and $D_s$ selection mechanism.

## 9 REPRODUCIBILITY STATEMENT

We provide code in both notebooks and python scripts. Notebooks consist of the dataset creation, coreset methods for selecting $D_s$ and ablation studies. Python scripts consist of the unlearning and evaluation. We provide a config file, which helps in configuring the settings for the Unlearning. Our anonymized code can be found at - link to the repo

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

# A APPENDIX

## A.1 DETAILS ON THE DATA SET CREATION

For the monotonic unlearning setting, we employed the Wikipedia Person Unlearn (WPU) dataset (Liu et al., 2024b), which contains 2,302 samples, including 10 designated samples corresponding to two entities that are intended to be forgotten. The WPU dataset was originally proposed for entity-level unlearning and is therefore well aligned with our objective.

To simulate a more realistic downstream scenario, we further constructed a mixed dataset by combining WPU with the TOFU dataset (Maini et al., 2024) and the Dolly dataset (Ouyang et al., 2022). This mixture serves two purposes: Dolly contributes general-purpose question–answering

Table 2: Num epochs for each $D_s$ on WPU and Mix datasets across coreset sizes.

| Coreset | 5% | | 10% | | 20% | | Full Retain | |
| | WPU | Mix | WPU | Mix | WPU | Mix | WPU | Mix |
| --- | --- | --- | --- | --- | --- | --- | --- | --- |
| random | 25 | 4 | 15 | 4 | 10 | 4 | – | – |
| MODERATE | 25 | 5 | 30 | 5 | 25 | 5 | – | – |
| EL2N (CE) | 30 | 5 | 25 | 5 | 25 | 5 | – | – |
| EL2N (diff) | 40 | 5 | 60 | 5 | 55 | 5 | – | – |
| Semantic | 30 | 5 | 15 | 5 | 15 | 5 | – | – |
| Syntactic | 20 | 5 | 15 | 5 | 15 | 5 | – | – |
| Full Retain | – | – | – | – | – | – | 8 | 2 |

data, while WPU introduces sensitive information that needs to be unlearned. Prior to merging, we assigned unique identifiers to each sample, derived from the dataset and entity of origin, to maintain traceability. We also incorporated the extended version of WPU introduced by Bushipaka et al. (2025), which includes indirect neighbor samples and a predefined test set. For computational tractability, samples exceeding 512 tokens were discarded. The final combined corpus consisted of approximately 21k samples: 14.2k from Dolly, 4k from TOFU, and 2.8k from WPU. From WPU, we selected 20 entities (98 samples) as the forgetting set.

Evaluating model utility on the entire 21k-sample corpus would have been computationally prohibitive. Instead, we curated a balanced test set. Specifically, for WPU we adopted the test partition provided by Bushipaka et al. (2025). From TOFU, we randomly sampled 500 instances, while for Dolly we applied stratified sampling across categories to preserve distributional diversity. This resulted in a test set comprising 1,992 samples in total.

## A.2 HYPERPARAMETERS FOR FINE-TUNING & UNLEARNING

For both Fine-tuning and Unlearning, we use LoRA Hu et al. (2022) since full fine-tuning and full unlearning is computationally expensive. For Fine-tuning, we used a batch size of 32, learning rate of 2e-5, LoRA rank=64, alpha =64 and for 10 epochs. Where as for Unlearning, we used a fixed batch size of 8, learning rate 1e-5, rank = 8, alpha = 16 for all the experiments. We used epochs as an hyperparameter (Table:2) to reach the FQ threshold of 0.90.

## A.3 UNLEARNING ALGORITHMS

### A.3.1 GRADIENT DIFFERENCE

Proposed by Liu et al. (2022) to mitigate the issues of Gradient ascent. It builds on the concept of Gradient Ascent, but not only aims to maximize the loss on forget set $D_f$, simultaneously minimizes the loss on the retain set $D_r$. This maintains the balance of forgetting and retaining. The loss function can be written as in equation 1.

Given $D$ and its samples $(x, y)$, $x$ is question and $y$ is the answer. A pair $p_i = p(x_i, y_i) \in D$ and $y_1, .. y_T$ are the answer tokens, we calculate Negative-Log-Likelihood (NLL) loss for $p_i$

$$\mathcal{L}(y \mid x; \theta) = \text{NLL}(y \mid x; \theta) = -\sum_{t=1}^{T} \log p(y_t \mid x, y_{<t}; \theta) \qquad (2)$$

**Gradient Ascent**'s main idea is to maximize the loss as opposed to the training objective of minimization by negating the loss. We can write it as

$$\mathcal{L}_{GA}(D_f; \theta) = -\mathcal{L}(y_f \mid x_f; \theta) \qquad (3)$$

From eq 1 and eq 3 we can write **Gradient Difference** as:

$$\mathcal{L}_{GD}(\theta) = -\mathcal{L}(D_f; \theta) + \mathcal{L}(D_r; \theta) \tag{4}$$

### A.3.2 SIMPLE NEGATIVE PREFERENCE

A modified variant of Negative Preference Optimization (NPO) (Zhang et al., 2024) that retains its core forgetting behavior by replacing the reference model with $\delta$ in the loss formulation of NPO.

NPO which is already an optimized algorithm of DPO made for unlearning eliminates the use of positive samples (often they are "IDK" samples), can be written as:

$$\mathcal{L}_{NPO,\beta}(\theta) = -\frac{2}{\beta} \, \mathbb{E}_{D_f} \left[ \log \sigma \Big( -\beta \log \frac{p(y \mid x; \theta)}{p(y \mid x; \theta_{\text{ref}})} \Big) \right] \tag{5}$$

SimNPO is optimized version of NPO, removes the need for reference model $\theta_{ref}$.

$$\mathcal{L}_{\text{SimNPO+retain}} = -\frac{2}{\beta} \, \mathbb{E}_{D_f} \left[ \log \sigma \Big( -\frac{\beta}{|y_f|} \log p(y \mid x; \theta) - \delta \Big) \right] + \gamma \, \mathcal{L}(D_r; \theta) \tag{6}$$

### A.3.3 REPRESENTATION MISDIRECTION UNLEARNING

RMU (Li et al., 2024) assumes knowledge is encoded in model parameters and manipulates these representations to suppress memorization signals for the forget set while preserving knowledge in the retain set. Let $\phi(s; \theta)$ denote the embedding features of the model, the loss is given by

$$\mathcal{L}_{\text{RMU+retain}} = \mathbb{E}_{D_f} \frac{1}{|y_f|} \sum_{i=1}^{|y_f|} ||\phi([x, y^{<i}]; \theta) - \text{c.u}||_2^2 + \mathcal{L}(D_r; \theta) \tag{7}$$

where u has elements randomly sampled from [0,1] and c is a scaling hyper-parameter.

## A.4 DETAILS ON THE DATA SELECTION METHODS

### A.4.1 MODERATE

The moderate coreset selection strategy was originally introduced in the context of classification tasks, wherein samples are partitioned into clusters according to their class labels. Since, our setting is instruction tuning, class labels are unavailable. We take the last token penultimate-layer representations from the pre-unlearned model for the full retain set (excluding $D_f$). We then partition it into four clusters using K-means algorithm. For each cluster, we determine its centroid and rank samples according to their distance from this centroid. To identify representative points, we select those whose distances are closest to the median within their respective clusters.

### A.4.2 EL2N

The central idea of the EL2N method is that the importance of each forget sample $z_f$ is quantified by the expected early-learning loss, measured as the $\ell_2$-norm of the model's prediction error during the early stages of training. Accordingly, the EL2N score for a forget sample is defined as

$$\chi(z_f) = \mathbb{E}_{\theta_t} \big\| f_{\theta_t}(x_f) - y_f \big\|_2, \quad \text{where } z_f = (x_f, y_f) \sim \mathcal{D}_f. \tag{8}$$

Here, the expectation is taken over snapshots of the model parameters $\theta_t$ from the early optimization trajectory, which captures how easily each sample is learned. Samples with higher EL2N scores correspond to those the model struggles to fit early on, and are therefore considered more influential. In practice, we compute this expectation over the unlearning trajectory, approximating it using 2 epochs for the monotonic setting and 1 epoch for the mixed setting in our experiments. As in our unlearning pipeline, this early trajectory includes a warmup phase based on the cross-entropy loss and also with Unlearning loss.

### A.4.3 SEMANTIC SIMILARITY

To calculate semantically closest samples to the forget samples, we used SBERT models. *all-MiniLM-L6-V2* (Reimers & Gurevych, 2019) for WPU dataset and *bge-small-en-v1.5* (Xiao et al., 2023) for Mix dataset. Then we picked the top semantically close samples to each forget sample by allocating a certain sample size for each until globally we reach the desired $D_s$ length. We didn't remove the duplicates cause multiple forget samples can be semantically closer to a few retain samples and increase in variance of the samples leads to low Model Utility (look into section6).

### A.4.4 SYNTACTIC SIMILARITY

In the light of recent analysis (Chang & Lee, 2025), that syntactic similarity is the most impacted by LLM Unlearning, we opted to do pick top synatctically similar samples as a $D_s$. To assess syntactic similarity between the forget and retain sets, each text was transformed into a sequence of part-of-speech tags and pairwise distances were computed using the normalized edit distance. This metric provides a principled quantification of structural correspondence, enabling a fine-grained comparison of syntactic patterns across the two sets (Zhang et al., 2017). Similar to Semantic, we allocate a certain sample size for each forget sample and incrementally increase it until we globally reach the desired $D_s$ length and we do not remove the duplicates.

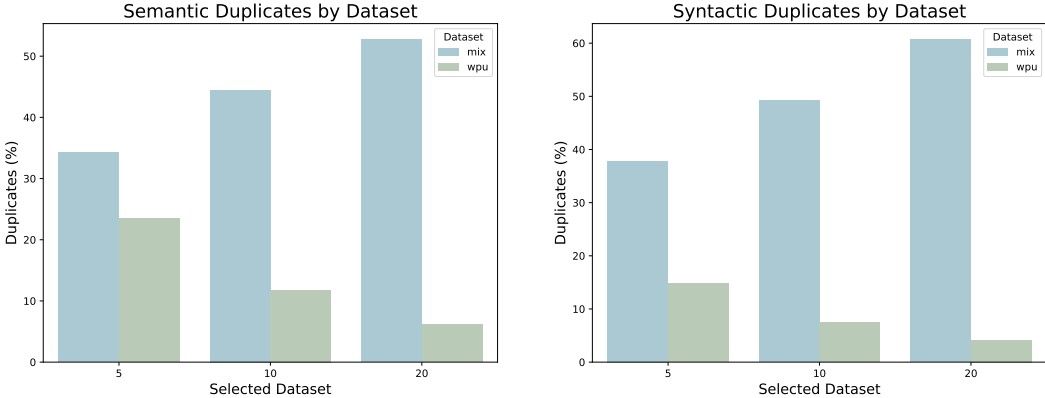

Figure 4: Percentage of duplicates in the Semantic and Syntactic $D_s$. As the $D_s$ size for Mix grows, duplicates increases, whereas in WPU we find the opposite trend.

### A.4.5 COMPUTATIONAL COSTS OF THE DATA SELECTION

From our empirical analysis, Random emerges as the least computationally expensive selection strategy, whereas EL2N with Gradient Difference incurs the highest cost (fig: 5). We quantify this cost as the total time required to identify the subset $D_s$, accounting for all pre-selection operations specific to each method. For instance, EL2N necessitates a warm-up phase, while syntactic selection requires part-of-speech tagging. Summing these pre-processing components provides a fair measure of computational overhead across methods.

Interestingly, Semantic selection ranks among the most efficient approaches—second only to Random—requiring only 30 seconds on the Mix dataset. Moreover, recent advances in semantic retrieval, particularly those leveraging vector databases, have made these methods increasingly practical and easier to implement compared to coreset-based alternatives. By contrast, Syntactic selection is considerably more time-intensive due to its reliance on CPU-bound processing rather than GPU acceleration.

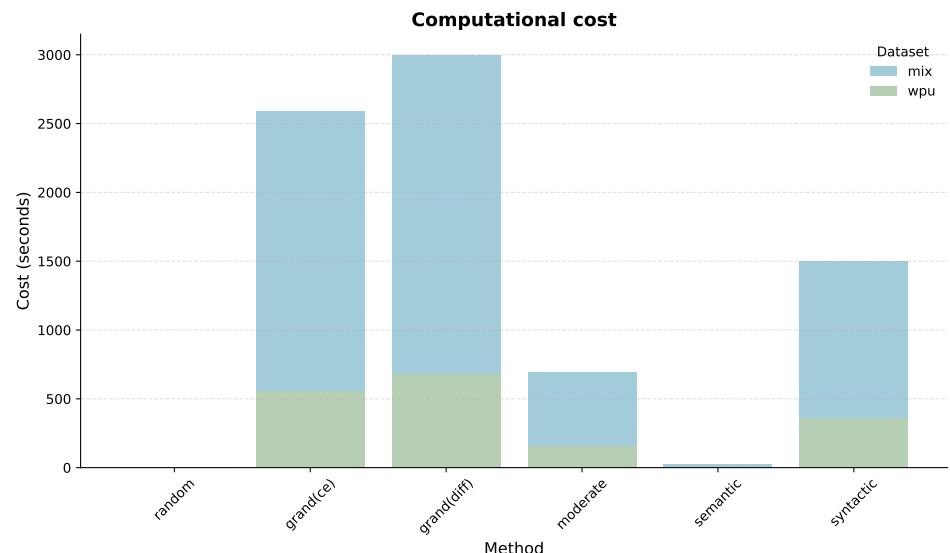

Figure 5: *Lower is better*, Time Taken for each method to select the $D_s$ in seconds. EL2N methods require significant amount of time compared to others. Note: EL2N is mispelled as GrAND here.

### A.5 EVALUATION METRICS

Following closely with (Maini et al., 2024; Yuan et al., 2025), we utilize a stack of metrics. All these scores are in range of $[0, 1]$.

#### A.5.1 ROUGE

We use ROUGE-L recall, which quantifies the model's output and the ground-truth answer. Given a generated response $g(x; \theta_*)$ and the ground-truth answer $y$, we employ $ROUGE - L(g(x; \theta_*), y)$.

#### A.5.2 PROBABILITY

Following (Maini et al., 2024) we compute the conditional probability $P(a|q)$ for the forget and retain sets and normalize the score by raising it to the power of $1/|a|$. Therefore the Probability can be written as $P(a|q)^{1/|a|}$.

#### A.5.3 COSINE-SIMILARITY

Provides the semantic similarity between $g(x; \theta_*)$ and $y$. Following (Yuan et al., 2025), we embed both the responses with a Sentence-BERT model (Reimers & Gurevych, 2019), and calculate the cosine-similarity between them. For evaluation, we used gte-small (Li et al., 2023). To keep the scores in $[0, 1]$, we truncate the values less than 0. It can be written as

$$\max\big(\cos\big(g(x; \theta_*), y\big), 0\big)$$

#### A.5.4 TRUTH RATIO

Introduced by (Maini et al., 2024), is often used to compute a ratio comparing the likelihood of the correct answer to incorrect ones. As stated in their work, since fine-tuning may inflate the probability of the exact ground-truth phrasing, they suggest to use a paraphrased version of the $y$ and average probabilities over multiple similarly formatted wrong answer. Let $\tilde{a}$ is the paraphrased answer and $\mathcal{A}_{pert}$ denote a set of five perturbed answers generated by GPT-4o. The truth ratio $\mathcal{R}_{truth}$ is calculated as:

$$R_{\text{truth}} = \frac{\frac{1}{|\mathcal{A}_{\text{pert}}|} \sum_{\hat{a} \in \mathcal{A}_{\text{pert}}} P(\hat{a} \mid q)^{q/|\hat{a}|}}{P(\tilde{a} \mid q)^{q/|\tilde{a}|}}$$

where $\mathcal{A}_{\text{pert}}$ is the perturbed answer set.

### A.5.5 RELATIVE MODEL UTILITY

Introduced by Chang & Lee (2025) to understand the behaviour of neighbor sets. It is a simple ratio to calculate the Utility drop pre-unlearning and post-unlearning.

$$RelativeUtilityDrop = \frac{MU_{pre} - MU_{post}}{MU_{pre}} \times 100 \tag{9}$$

## A.6 RESULTS

### A.6.1 FORGET QUALITY AND MODEL UTILITY

Table 3: Forget Quality and Model Utility for WPU and Mixed Datasets

| Coreset (→) | FQ ↑ | | | MU ↑ | | |
|---|---|---|---|---|---|---|
| | 5 | 10 | 20 | 5 | 10 | 20 |
| **WPU Dataset** | | | | | | |
| random | 0.94 | 0.94 | 0.94 | 0.35 | 0.37 | 0.56 |
| MODERATE | 0.94 | 0.95 | 0.96 | 0.27 | 0.58 | 0.74 |
| EL2N (CE) | 0.90 | 0.95 | 0.93 | 0.21 | 0.43 | 0.68 |
| EL2N (diff) | 0.95 | 0.95 | 0.95 | 0.30 | 0.64 | **0.79** |
| Semantic | 0.96 | 0.93 | 0.93 | 0.32 | 0.64 | 0.76 |
| Syntactic | 0.92 | 0.93 | 0.95 | 0.23 | 0.61 | **0.79** |
| **Mix Dataset** | | | | | | |
| random | 0.92 | 0.93 | 0.93 | 0.59 | 0.50 | 0.55 |
| MODERATE | 0.94 | 0.93 | 0.93 | 0.41 | 0.49 | 0.58 |
| EL2N (diff) | 0.94 | 0.95 | 0.93 | 0.21 | 0.40 | 0.50 |
| EL2N (CE) | 0.94 | 0.94 | 0.93 | 0.13 | 0.35 | 0.46 |
| Semantic | 0.94 | 0.91 | 0.92 | 0.21 | 0.59 | **0.66** |
| Syntactic | 0.93 | 0.94 | 0.94 | 0.17 | 0.35 | 0.63 |

### A.6.2 RELATIVE UTILITY DROP FOR CORESET METHODS

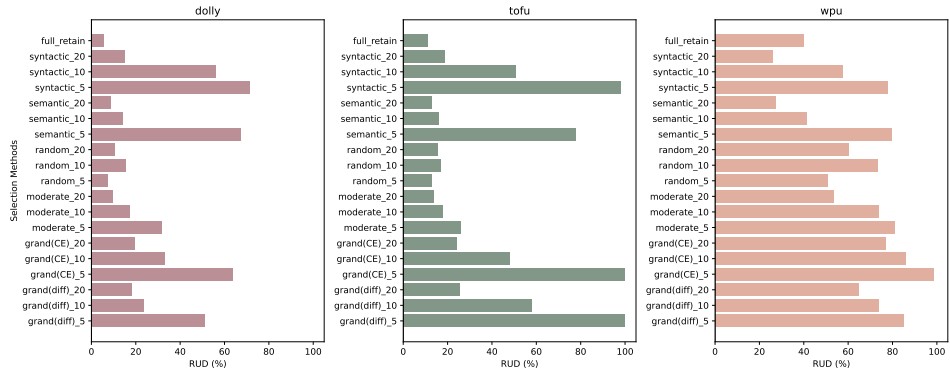

Figure 6: *Lower is better*, Relative Utility Drop (RUD) on the Mixed Test dataset across all the different sources. WPU has the highest and Dolly has the lowest RUD across all the settings. Note: EL2N is misspelled as GrAND here.

Table 4: Relative Utility Drop (↓%) on WPU and Mix datasets across coreset sizes for Gradient Diff, SimNPO, and RMU. Lower is better.

| Coreset | 5% | | 10% | | 20% | |
|---|---|---|---|---|---|---|
| | WPU | Mix | WPU | Mix | WPU | Mix |
| **GradDiff** | | | | | | |
| random | 63.92 | 21.33 | 62.89 | 33.33 | 42.27 | 26.67 |
| MODERATE | 72.16 | 45.33 | 40.21 | 34.67 | 24.74 | 22.67 |
| EL2N (CE) | 78.35 | 53.33 | 55.58 | 28 | 31.96 | 14.67 |
| EL2N (diff) | 59.59 | 74.67 | 35.05 | 28 | 18.56 | **14.6** |
| Semantic | 67 | 72 | 34.02 | 21.33 | 21.65 | **12** |
| Syntactic | 76.29 | 77.33 | 37.11 | 53.33 | **18.56** | 16 |
| **SimNPO** | | | | | | |
| random | 11.34 | 25.33 | 10.31 | 22.67 | **8.25** | 13.33 |
| MODERATE | 15.46 | 25.2 | 12.37 | 24 | **8.25** | 14.67 |
| EL2N (CE) | 23.71 | 28 | 18.56 | 26.67 | 11.34 | 22.67 |
| EL2N (SimNPO) | 14.13 | 32 | 12.37 | 30.67 | 10.31 | 28.0 |
| Semantic | 16.49 | 20 | 12.37 | 18.67 | 9.28 | **13.33** |
| Syntactic | 16.49 | 18.67 | 10.31 | 17.33 | **8.25** | 14.67 |
| **RMU** | | | | | | |
| random | 59.79 | 62.67 | 40.21 | 48 | 48.45 | 42 |
| MODERATE | 68.04 | 62.5 | 41.24 | 49.33 | 42.27 | 41.33 |
| EL2N (CE) | 65.98 | 76 | 49.48 | 64 | 51.55 | 49.33 |
| EL2N (RMU) | 75.26 | 69.33 | 54.64 | 61.33 | 43.3 | 48 |
| Semantic | 74.23 | 68 | 50.52 | 41.33 | 43.3 | **37.33** |
| Syntactic | 75.26 | 76 | 42.27 | 45.33 | 42.26 | 38.67 |

In the main sections of the paper, we provided *RUD*, we report all the results in Table:3 that include *Forget Quality* and *Model Utility*. As mentioned in previously in section:5, we made sure all the Unlearning experiments crossed the threshold of FQ $> 0.90$. The *FQ* ranges from 0.90-0.95. We also provide per source RUD scores for the Mix dataset. We find that WPU is the most impacted and DOLLY is the least impacted. Given that all our forget samples are from WPU, this can be expected.

### A.6.3    CLUSTERING

Since investigating and finding relations between every pair is an NP-hard problem, we approach this with clustering the HSV representations to $k = 10$ clusters with k-means algorithm. We chose $k = 10$ based on the elbow method. We find that best performing methods select samples mostly from clusters 9 and 2 (for Mix). A strange behavior is from Random (on mix), which selects almost uniformly from all the clusters. Although small, Random 5 outperforms 10 and 20 (Mix). However this selection needs to be studied more.

### A.6.4    LOG PREFERENCE RATIO

To analyze how different selection strategies distribute their retain sets across the representation space, we introduce the *preference ratio*. For each cluster $c$, we compute the retain cluster share

$$p_{\text{retain}}(c) = \frac{\text{retain\_count}(c)}{\text{retain\_total}},$$

and compare it to the baseline cluster share in the non-forget pool

$$q_{\text{pool}}(c) = \frac{\text{pool\_count}(c)}{\text{pool\_total}}.$$

The preference ratio is then defined as

$$\text{pref\_ratio}(c) = \frac{p_{\text{retain}}(c)}{q_{\text{pool}}(c)}.$$

To improve interpretability, we report results in logarithmic scale:

$$\text{pref\_log2}(c) = \log_2\left(\text{pref\_ratio}(c)\right).$$

Here, $\text{pref\_ratio}(c) > 1$ indicates that the method oversamples cluster $c$, $\text{pref\_ratio}(c) < 1$ indicates undersampling, and $\text{pref\_log2}(c) = 0$ denotes neutral selection. This formulation allows us to visualize selection biases at the cluster level and to relate them to model utility and forgetting efficacy.

### A.6.5 CORRELATIONS

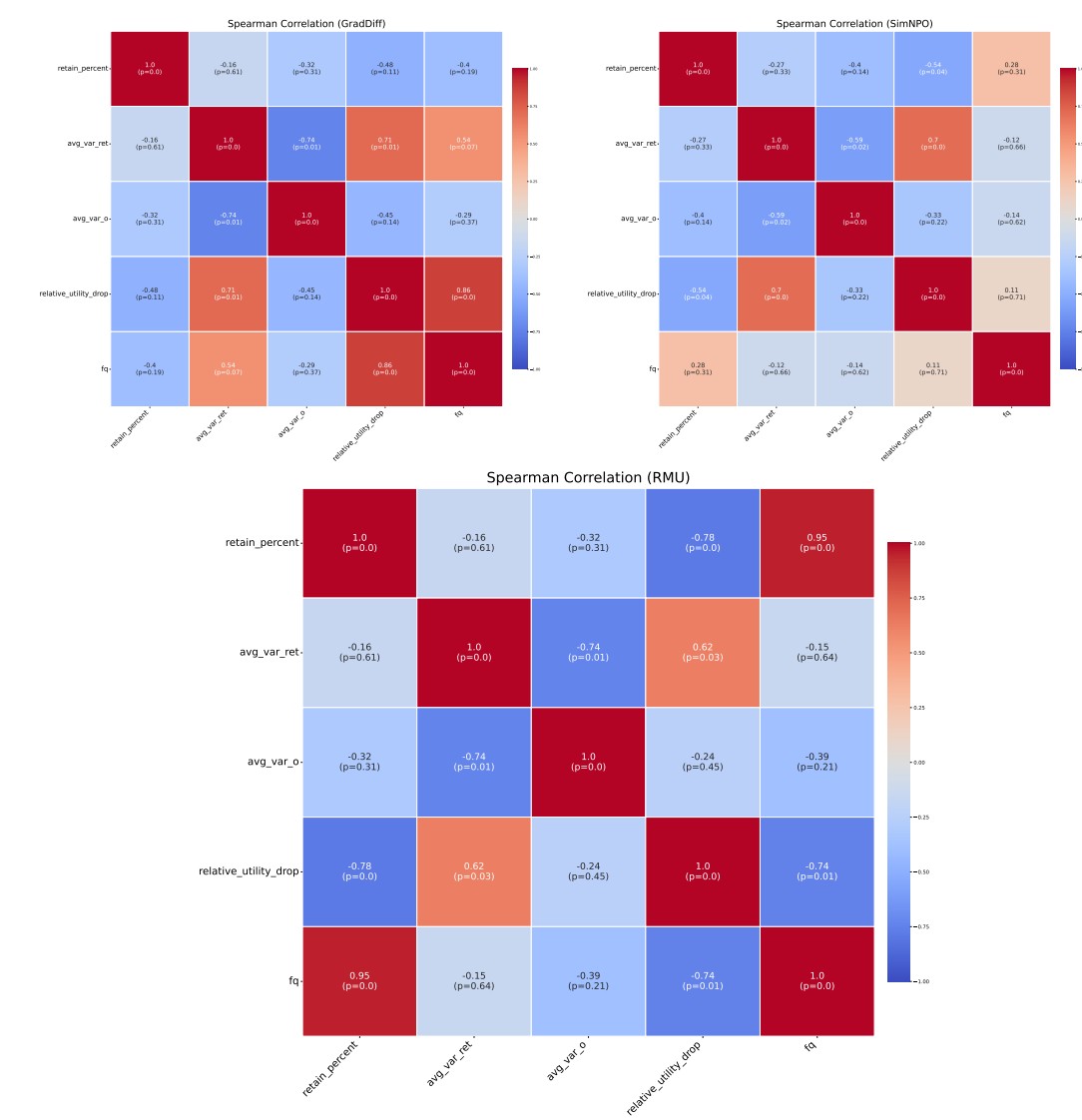

Figure 7: The correlations of $Var(D_s)$ data points with RUD and FQ for all the algorithms.

## A.7 MMLU SCORES

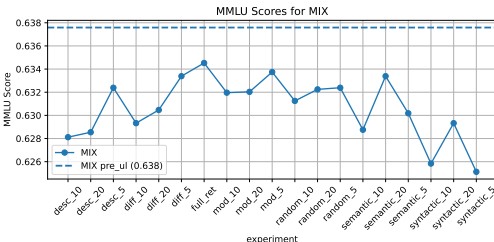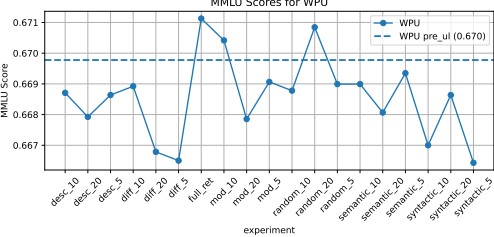

Figure 8: MMLU scores for GradDiff experiment. We do not find significant deviation in the scores of MMLU post-unlearning for all the methods.

## A.8 CAUSALITY LINK

| Unlearning Method | Low | | Medium | | High | |
|---|---|---|---|---|---|---|
| | **FQ** | **MU** | **FQ** | **MU** | **FQ** | **MU** |
| GradDiff | 0.94 | 0.51 | 0.93 | **0.58** | 0.93 | 0.57 |
| SimNPO | 0.87 | 0.51 | 0.85 | **0.54** | 0.84 | 0.54 |
| RMU | 0.91 | 0.34 | 0.92 | **0.38** | 0.92 | 0.35 |

Table 5: Comparison across unlearning methods with low, medium, and high difficulty (FQ and MU).

## A.9 LLM USAGE

In our study, we utilized LLMs for polishing the writing, research paper gathering, and coding. We used LLM to polish writing in all the sections of the paper, however we made sure it didn't hallucinate and add made up information. In the initial stages of our study, we used deep research tool for research papers gathering on coresets. For mix dataset construction, investigations, and parts of Unlearning we used LLM for coding.

