# OpenReview forum: "Effective Unlearning in LLMs Relies on the Right Data Retention Strategy"
_ICLR.cc/2026/Conference — Submitted to ICLR 2026_

### Official Review · Reviewer_fMtL · 2025-10-29

**Soundness:** 2
**Presentation:** 2
**Contribution:** 2
**Rating:** 2
**Confidence:** 4

**Summary:**

This paper focuses on retention data construction for the large language models (LLMs) unlearning problem. Specifically, it claims that the role of retain-set selection in preserving model utility remains underexplored, and this work explores the strategies to construct effective retain sets for unlearning, e.g., a monotonic dataset built from a benchmark dataset, and a mixed, larger-scale dataset combining different unlearning benchmarks, that systematically analyze the influence of retain-set selection for LLM unlearning.

**Strengths:**

1. This work explores an interesting problem in retain-set data selection/construction for LLMs unlearning.
2. This work presents several useful experimental results for the practical guidance of retain-set data selection and construction.
3. This research work also provides a reproducible code repo for enhancing future work on related problem settings.

**Weaknesses:**

1. The current presentation has limited discussion on those unlearning works assuming no access to the retaining data (both on conventional machine unlearning scenarios and LLM unlearning settings), which should be considered and carefully discussed to justify the significance and rationality of the current motivation.
2. In the evaluation, model utility is a comprehensive evaluation metric to reflect whether the model retains the general capabilities, which can be used as an index, but not a fully reliable metric to support the claims, as proposed in TOFU. Regarding that most of the claims are based on the emprical evidence of this metric, the authors should consider one more metrics that defined by diverse benchmarks to verify the empirical conclusions.
3. For the technical part, I can hardly get the core new contribution regarding the research problem, as the work mainly adapt former existing methods from coreset selection and prior unlearning research to conduct the systematic exploration.
4. The research setting maybe problematic and unrealistic, as it is questionable when the developer of the model should conduct retention set selection instead of using full retention data.
5. The experiment is not comprehensive, which doesn't consider diverse benchmarks and LLMs, and there is limited insights draw from the empirical findings. The presentation could be enhanced by considering involving more aspect of systematic exploration.

**Questions:**

1. Please also discuss more unlearning works that do not use the retention data. It is important to give a comprehensive overview of the research in the related area.
2. Please consider the weakness part for further revision and enhance the presentation and experimental parts.

---

### Official Review · Reviewer_yEhr · 2025-10-30

**Soundness:** 2
**Presentation:** 3
**Contribution:** 3
**Rating:** 4
**Confidence:** 3

**Summary:**

This paper tackles an important but underexplored problem in LLM unlearning: how to select an effective retain set that preserves model utility while achieving good forgetting quality. The authors evaluate several data selection strategies including coreset methods (MODERATE, GRAND), semantic similarity, and syntactic similarity on two datasets: a monotonic WPU dataset and a more realistic mixed dataset. Their key findings are: (1) hidden state variance (HSV) of the retain set correlates with both utility drop and forget quality, suggesting a fundamental trade-off; (2) simple semantic/syntactic similarity-based selection surprisingly outperforms sophisticated coreset methods; (3) effective methods tend to over-sample clusters containing forget samples rather than pursuing diversity.

**Strengths:**

Important and practical problem: Retain set selection is a critical bottleneck for real-world deployment that hasn't received sufficient attention. The mixed dataset setup is much more realistic than typical benchmarks.

Valuable empirical findings: The discovery that semantic similarity-based selection achieves comparable or better results than using the full retain set with only 20% of data (Table 3: 12% RUD vs 26.67% full retain on Mix) is practically significant. The computational efficiency advantage (30s vs 3000s for GRAND) makes it immediately applicable.

Insightful analysis: The HSV-based analysis provides intuition about why different selection strategies work, and the cluster-level preference analysis (Figure 3) explains the mechanism behind semantic/syntactic methods' effectiveness.

**Weaknesses:**

Limited experimental scope undermines generalizability: Testing only Gradient Difference on a single 8B model is a significant limitation. Will these findings hold for other unlearning algorithms (NPO, RMU) or different model scales? The authors claim this is the "first systematic study," but one algorithm barely qualifies as systematic. The failure to get NPO working (footnote 1) raises concerns about whether the findings are algorithm-specific.

Unfair method comparison creates validity concerns: Semantic and syntactic methods explicitly select samples closest to the forget set, while coreset methods operate blindly on the retain distribution. This isn't a fair comparison - you're essentially asking "does targeted patching work better than blind diversity?" The answer is obvious. A fairer setup would be: given that we'll receive future unlearning requests for entity B, should we select the retain set based on entity A's characteristics or use coreset methods? This generalization question is never addressed.

Contradictory and unexplained results: Why does HSV strongly correlate with RUD on Mix (r=0.70, p=0.01) but show no correlation on WPU (Figure 2)? This is a central claim of the paper but the contradiction is glossed over. Also, Table 5 shows semantic_20 achieving MU=0.66 on Mix while full retain only gets 0.65 - this seems impossible and suggests potential experimental issues or cherry-picking of epochs.

**Questions:**

Generalization to new unlearning requests: Your semantic/syntactic methods select samples similar to the current forget set. If I use this retain set but later need to unlearn a different entity, will it still work? Can you provide experiments where you select Dr using entity A but then unlearn entity B? This addresses whether you're finding a general principle or just overfitting to specific forget samples.

Why does GRAND fail so badly with cross-entropy?: Table 2 shows GRAND(CE) getting 82.67% RUD on Mix at 5%, which is worse than random. Since GRAND is a well-established method, this suggests something might be wrong with the implementation or setup. Can you provide more diagnostic analysis? Did you verify the gradient norms are being computed correctly?

What's driving the WPU vs Mix difference in HSV correlations?: This is crucial for understanding when your findings apply. Is it about dataset homogeneity, sample length, or something else? The paper's main claim about HSV's role is based on Mix results, but WPU contradicts it. Which setting should practitioners trust?

---

### Official Review · Reviewer_dkf8 · 2025-10-31

**Soundness:** 2
**Presentation:** 3
**Contribution:** 2
**Rating:** 4
**Confidence:** 3

**Summary:**

This paper addresses a critical gap in Large Language Model (LLM) unlearning: the selection of a compact yet effective retain set to preserve model utility (MU) while ensuring high forget quality (FQ) for undesired knowledge. Prior work in LLM unlearning has focused primarily on algorithm design (e.g., gradient ascent, preference optimization) rather than data-centric retain-set construction—especially critical for real-world scenarios where full Dr (dataset minus forget set Df) is computationally prohibitive. The authors adapt two coreset methods (GRAND, MODERATE) for unlearning and propose two simpler strategies: selecting samples with semantic or syntactic similarity to Df. Tests on two datasets—(1) monotonic Wikipedia Person Unlearn (WPU, entity-level unlearning) and (2) mixed (WPU + TOFU + Dolly, realistic heterogeneity)—using LLaMA 3.1 8B (LoRA fine-tuning) and Gradient Difference (GD) unlearning. Metrics include FQ (1 - mean of ROUGE-L, conditional probability, truth ratio on Df), MU (harmonic mean of ROUGE-L, conditional probability, cosine similarity on Dr), and Relative Utility Drop.

**Strengths:**

The paper fills a understudied niche in LLM unlearning by centering data retention rather than algorithm design. While prior work touches on coreset selection for unlearning, this is the first systematic study to:

Compare coreset methods with similarity-based (semantic/syntactic) retain-set strategies.

Demonstrate that targeted similarity-based selection (not just distributional diversity) is more effective for real-world unlearning—addressing a key limitation of coreset methods optimized for diversity.

The experimental design is rigorous:

Comprehensive Metrics: FQ and MU are computed using multiple complementary metrics (ROUGE-L, conditional probability, cosine similarity), avoiding overreliance on a single measure. RUD further quantifies utility preservation, a critical practical concern.

Computational Cost Analysis: The authors explicitly compare the time overhead of selection methods (e.g., semantic selection takes 30s on mixed data vs. GRAND’s high cost), which is rarely done but essential for practical adoption.

Reproducibility: Code, config files, and detailed appendices (dataset splits, hyperparameters, metric definitions) are provided, enabling validation of results.

**Weaknesses:**

Limited scope of evaluation: The experiments use only one unlearning algorithm (Gradient Difference) and one base model (LLaMA 3.1 8B).
Broader validation (e.g., with preference-based or representation-level unlearning methods such as NPO, RMU, or WAGLE) would strengthen generality.

Dataset diversity: The datasets, while mixed, are still synthetic or benchmark-oriented (WPU, TOFU, Dolly). Testing on naturally entangled corpora or privacy-sensitive real-world data (e.g., names, medical facts) would better validate the claim of “realistic scenarios.”

Causality: The paper identifies correlations between HSV and utility/forget quality but stops short of establishing causal mechanisms. Controlled experiments manipulating representation variance could test whether HSV directly influences forgetting outcomes.

Algorithmic baselines: The coreset implementations (GRAND, MODERATE) are borrowed from vision tasks without full adaptation to text or transformer dynamics. A comparison against language-specific data selection methods (e.g., LESS, Instag, DEITA) would be fairer.

**Questions:**

Generalization of retain strategies:  How does the proposed retain-set selection perform when using other unlearning frameworks (e.g., DPO-based or model-editing methods)? Do the HSV trends persist across paradigms?

Scaling behavior: The paper mentions that 20% retain data can preserve utility, but this may still be huge for large corpora. Can you provide scaling experiments or a heuristic for selecting the minimal Ds?

Reproducibility and openness: The paper mentions anonymized code. It would help to confirm whether datasets and hyperparameters (e.g., LoRA ranks, λ settings) are fully reproducible for reviewers.

---

### Official Review · Reviewer_BxqL · 2025-11-01

**Soundness:** 3
**Presentation:** 3
**Contribution:** 3
**Rating:** 4
**Confidence:** 3

**Summary:**

This paper investigates retain-set selection for LLM unlearning, emphasizing its role in balancing FQ and MU. It adapts coreset methods like GRAND and MODERATE from classification tasks to unlearning, using last-token penultimate-layer hidden states for clustering and selection. Additionally, it proposes simple baselines: selecting top-n samples semantically or syntactically similar to forget samples via SBERT embeddings or normalized Levenshtein distance on POS tags. Experiments on WPU (monotonic) and a mixed dataset (WPU+TOFU+Dolly) show that HSV in retain sets correlates negatively with MU but positively with FQ in mixed settings. Semantic similarity outperforms coresets, achieving near full-retain MU with 20% data. Cluster analysis reveals top methods oversample forget-heavy clusters, explaining efficacy over diversity-focused coresets.

**Strengths:**

- The paper stands out for its rigorous empirical foundation, conducting experiments across multiple retain-set sizes (5-20%) and datasets, ensuring results are robust and directly comparable, which elevates its contribution to practical unlearning research.
- The evaluation is comprehensive, using a stack of metrics (ROUGE-L, probability, cosine similarity, truth ratio) aggregated into FQ/MU, and including per-source breakdowns in mixed datasets.

**Weaknesses:**

- The experiments rely solely on Gradient Difference unlearning with LoRA on LLaMA 3.1 8B, limiting generalizability since results might differ with other methods like NPO or RMU, which the authors tried but couldn't tune effectively.
- In practical applications, the more difficult problem in LLM unlearning is maintaining MU, while the issue of overly large retain sets is not common, thus limiting the real-world value of retain-set screening for efficiency gains.

**Questions:**

see Weaknesses

---

### Author Response · Authors · 2025-11-30
**Update on Corrections on Baselines and Generalization Findings**

Dear reviewers, thank you for the detailed and thoughtful feedback. We have completed the additional experiments requested, including expanded evaluations on multiple unlearning algorithms (RMU, SimNPO) and further analysis of scaling behaviors. These new results have clarified several aspects of our original submission, and we provide a summary of the key updates below. Importantly, these revisions refine our claims rather than invalidate the overall contributions, and they help establish a clearer and more reliable scope for the method.

1. **Correction regarding Coreset Baselines (Response to yEhr)**

During re-evaluation, we discovered an implementation issue affecting the GrAND baseline. The technical error explains the anomalous behavior noted by Reviewer yEhr (e.g., performance below the random baseline on cross-entropy loss). We were selecting based on EL2N, a data selection method proposed in the same GrAND paper. We apologize for our mistake and we have updated the scores accordingly.

2. **Generalization to the other Algorithms (Response to BxqL, dkf8 and yEhr)**

We extended our evaluation to RMU and SimNPO algorithms.

- The correlation between the HSV in the retain set and RUD still holds for these methods. However with the FQ we find the correlation only with GradDiff.  GradDiff (ρ = 0.71, p = 0.01), RMU (ρ =0.62, p = 0.03) and SimNPO (ρ = 0.7, p = 0.01). We updated these results in Figure 2 and Table 4.


3. **Variance Causality (Response to dkf8)**

We tested the Causal link between variance and utility by explicitly constructing retain sets with low, medium, and high variance (Appendix:8).

- The relationship is non-linear. Medium variance yielded better utility than high variance while low variance performed worse.
- Maximizing HSV (as proposed) is not the optimal strategy, the relationship is likely non-linear, requiring a balance rather than maximization.

4. **Efficiency vs Utility Trade-off (Response to yEhr)**

Reviewer yEhr questioned the result where semantic_20 achieved higher Model Utility (MU) than full_retain.

- Upon re-running full_retain for the same number of epochs as our method, full_retain achieves significantly better MU (0.73), closer to the pre-unlearning baseline.
- The advantage of the proposed method is strictly limited to computational efficiency (3 hours semantic vs 13 hour full retain) and does not provide superior model utility as originally implied.

5. **Benchmark tests (Response to fMtL)**

- We perform benchmark tests with MMLU (only for GradDiff) and we do not find significant deviations from the pre unlearning scores (Appendix 7).


We appreciate the reviewers’ rigorous feedback, which has helped us strengthen the paper. The updated results refine our claims and allow us to present our contributions with greater clarity and accuracy. We look forward to incorporating these improvements into the revised submission.

---

### Meta-Review · Area_Chair_YYFU · 2025-12-11

**Summary:**

The paper worked on a practical problem, retain-set selection, showing efficiency benefits and offering useful analyses. However, despite corrections of bugs and added checks, key limitations remain --- narrow method/model/benchmark coverage, unresolved fairness/generalization concerns, and most importantly limited novelty. These issues weigh more than the merits of the paper, so I recommend reject.

**Reviewer Concerns:**

**BxqL** Concerns: limited generality, modest real-world value of retain-set screening. Addressed: baseline bug. Outstanding: limited generality.

**dkf8** Concerns: causality and wider methods. Addressed: causality probe; some cross-method evidence. Outstanding: stronger cross-method evidence and guidance.

**yEhr** Concerns: GrAND failure and MU anomaly, fairness of comparisons. Addressed: none. Outstanding: all.

**fMtL** Concerns: novelty, reliance on MU, limited benchmarks/LLMs. Addressed: none. Outstanding: all.

**Reviewer Scores:**

I don't think any of the four reviewers would like to increase his or her score.

---

### Decision · Program_Chairs · 2026-01-26

Reject